# Comparative 6-Month Wild-Type and Delta-Variant Antibody Levels and Surrogate Neutralization for Adults Vaccinated with BNT162b2 versus mRNA-1273

Brian Grunau,[a,b,c] Liam Golding,[d] Martin A. Prusinkiewicz,[e] Michael Asamoah-Boaheng,[b,f] Richard Armour,[c] Ana Citlali Marquez,[g,h] Agatha N. Jassem,[g,h] Vilte Barakauskas,[g] Sheila F. O'Brien,[i] Steven J. Drews,[i,j] Scott Haig,[c] Pascal M. Lavoie,[e] David M. Goldfarb[g]

aCentre for Health Evaluation & Outcome Sciences, University of British Columbia, Vancouver, British Columbia, Canada
bDepartment of Emergency Medicine, University of British Columbia, Vancouver, British Columbia, Canada
cBritish Columbia Emergency Health Services, Vancouver, British Columbia, Canada
dDepartment of Obstetrics and Gynecology, University of British Columbia, Vancouver, British Columbia, Canada
eDepartment of Pediatrics, University of British Columbia, Vancouver, British Columbia, Canada
fFaculty of Medicine, Clinical Epidemiology, Memorial University of Newfoundland, St. John's, Newfoundland, Canada
gDepartment of Pathology and Laboratory Medicine, University of British Columbia, Vancouver, British Columbia, Canada
hPublic Health Laboratory, British Columbia Centre for Disease Control, Vancouver, British Columbia, Canada
iCanadian Blood Services, Ottawa, Ontario, Canada
jLaboratory Medicine and Pathology, University of Alberta, Edmonton, Alberta, Canada

**ABSTRACT** While mRNA vaccines are highly efficacious against short-term COVID-19, long-term immunogenicity is less clear. We compared humoral immunogenicity between BNT162b2 and mRNA-1273 vaccines 6 months after the first vaccine dose, examining the wild-type strain and multiple Delta-variant lineages. Using samples from a prospective observational cohort study of adult paramedics, we included COVID-19-negative participants who received two BNT162b2 or mRNA-1273 vaccines, and provided a blood sample 170 to 190 days post first vaccine dose. We compared wild-type spike IgG concentrations using the Mann-Whitney U test. We also compared secondary outcomes of: receptor binding domain (RBD) wild-type antibody concentrations, and inhibition of angiotensin-converting enzyme 2 (ACE-2) binding to spike proteins from the wild-type strain and five Delta-variant lineages. We included 571 adults: 475 BNT162b2 (83%) and 96 mRNA-1273 (17%) vaccinees, with a mean age of 39 (SD = 10) and 43 (SD = 10) years, respectively. Spike IgG antibody concentrations were significantly higher ($P < 0.0001$) for those who received mRNA-1273 (GM 601 BAU/mL [GSD 2.05]) versus BNT162b2 (GM 375 BAU/mL [GSD 2.33]) vaccines. Results of RBD antibody comparisons ($P < 0.0001$), and inhibition of ACE-2 binding to the wild-type strain and all tested Delta lineages (all $P < 0.0001$), were consistent. Adults who received two doses of mRNA-1273 vaccines demonstrated improved wild-type and Delta variant-specific humoral immunity outcomes at 6 months compared with those who received two doses of the BNT162b2 vaccine.

**IMPORTANCE** The BNT162b2 and mRNA-1273 mRNA SARS-CoV-2 vaccines have demonstrated high efficacy for preventing short-term COVID-19. However, comparative long-term effectiveness is unclear, especially pertaining to the Delta variant. We tested virus-specific antibody responses 6 months after the first vaccine dose and compared individuals who received the BNT162b2 and mRNA-1273 SARS-CoV-2 vaccines. We found that individuals who received the mRNA-1273 vaccine demonstrated superior serological markers at 6 months in comparison with those who received the BNT162b2 vaccine.

**KEYWORDS** SARS-CoV-2, Delta, spike, COVID-19

Address correspondence to Brian Grunau, Brian.Grunau2@vch.ca.

The authors declare a conflict of interest. S.D. has acted as a content expert for respiratory viruses for Johnson & Johnson (Janssen).

Clinical trials have found the BNT162b2 and mRNA-1273 vaccines to be highly efficacious against short-term severe COVID-19 (1, 2). However, as clinical trial testing was performed against placebo, there is less clarity on the comparative long-term effectiveness of these two mRNA vaccines.

Existing clinical trial testing was performed when the prototypic wild-type strain was predominant (1, 2). The Delta variant (B.1.617.2) was first detected in late 2020, and spread rapidly, becoming the dominant global strain by mid-2021 (3). This variant was found to have an increased risk of hospitalization and death, in comparison to the wild-type strain (4). On December 14, 2021, the Delta variant accounted for 99.2% of COVID-19 cases globally (5). Delta strain mutations resulted in multiple Delta variant lineages with variable predominance throughout different regions of the world (6). While several studies have compared disease severity and immunogenicity of the Delta strain with other strains, differences in vaccine effectiveness among the individual Delta variant lineages is less clear (4, 7–9).

The comparative long-term immunogenicity conveyed from different mRNA vaccines, particularly in relation to the Delta variant is unclear. Further, it is unclear if vaccine-based immunogenicity differences are consistent among the multiple Delta variant lineages; or alternatively, whether differing vaccination strategies may be required depending on which Delta variant is most prevalent in a community. For these reasons, we compared humoral immunogenicity between BNT162b2 and mRNA-1273 vaccinees at 6 months after the first dose, examining the wild-type strain and several different Delta variant lineages.

## RESULTS

**Participant characteristics.** The study included 571 adults (enrolled January 22, 2021 to October 7, 2021); 475 (83%) and 96 (17%) received two doses of the BNT162b2 vaccine or two doses of the mRNA-1273 vaccine, respectively. BNT162b2 and mRNA-1273 vaccinees demonstrated similar mean age and sex distribution (Table 1). The average vaccine dosing interval was 7 weeks in both groups, with similar timing of doses and blood sampling.

**Primary outcome comparison.** Spike IgG antibody concentrations, measured on the V-PLEX assay (Fig. 1), were significantly higher ($P < 0.0001$) for mRNA-1273 (GM 601 BAU/mL [GSD 2.05]) versus BNT162b2 (GM 375 BAU/mL [GSD 2.33]) vaccinees.

**Secondary outcome comparisons.** Spike total antibody concentrations were significantly higher for mRNA-1273, in comparison with BNT162b2, vaccinees when measured on the Elecsys assay (GM 3223 BAU/mL [GSD 2.36] versus GM 1895 U/mL [GSD 3.12]; $P < 0.0001$). Receptor binding domain (RBD) antibody concentrations were significantly higher for mRNA-1273, in comparison with BNT162b2, vaccinees (GM 1815 BAU/mL [GSD 2.05] versus GM 1131 BAU/mL [GSD 2.33]; $P < 0.0001$).

Inhibition of ACE-2 binding was significantly higher for mRNA-1273, in comparison with BNT162b2 vaccinees, respectively, for the: wild-type strain (GM 14.13 U/mL [GSD 2.74] versus GM 8.06 U/mL [GSD, 3.03]; $P < 0.0001$); as well as for the AY.1 (GM 13.24 U/mL [GSD 2.47] versus GM 8.85 U/mL [GSD 2.76]; $P < 0.0001$), AY.2 (GM 12.21 U/mL [GSD 2.41] versus GM 8.43 U/mL [GSD, 2.67]; $P < 0.0001$), B.1.617.2/AY.3/AY.5/AY.6/AY.7/AY.14 (GM 13.95 U/mL [GSD 2.56] versus GM 8.97 U/mL [GSD, 2.86], $P < 0.0001$), B.1.617.2/AY.4 (GM 14.90 U/mL [GSD 2.39] versus GM 9.71 U/mL [GSD 2.77]; $P < 0.0001$), and AY.12 (GM 12.52 U/mL [GSD 2.52] versus GM 8.13 U/mL [GSD, 2.83] $P < 0.0001$) Delta variant spike proteins (Fig. 2).

## DISCUSSION

We examined SARS-CoV-2 surrogates of virus neutralization and antibody concentrations 6 months after the initial SARS-CoV-2 vaccine, specifically examining the wild-type and multiple Delta variant strains, among 571 middle-aged adults who received two doses an mRNA vaccine. We found immune measures to be significantly higher in mRNA-1273-vaccinated individuals, compared to the BNT162b2-vaccinated individuals with similar characteristics, regardless of the target virus strain tested.

**TABLE 1** Participant characteristics[a]

| Characteristics | BNT162b2 (n = 475) | mRNA-1273 (n = 96) |
|---|---|---|
| Age (yr), mean (SD)[b] | 39 (10) | 43 (10) |
| Female sex, n (%) | 219 (46) | 39 (41) |
| Vaccination | | |
| January 1, 2021 to 1st vaccine interval (d), mean (SD) | 16 (15) | 21 (19) |
| 1st vaccine to blood sample interval (d), mean (SD) | 182 (4.5) | 183 (4.5) |
| 2nd vaccine to blood sample interval (d), mean (SD) | 52 (34) | 52 (26) |
| Vaccine dosing interval (d), mean (SD) | 51 (37) | 52 (26) |
| Past medical history | | |
| Hypertension, n (%) | 31 (6.5) | 14 (15) |
| Diabetes, n (%) | 8 (1.7) | 2 (2.1) |
| Asthma, n (%) | 62 (13) | 15 (16) |
| Lung disease, n (%) | 4 (0.84) | 0 |
| Heart disease, n (%) | 3 (0.63) | 1 (1.0) |
| Kidney disease, n (%) | 1 (0.21) | 0 |
| Liver disease, n (%) | 7 (1.5) | 1 (1.0) |
| Cancer, n (%) | 9 (1.9) | 4 (4.2) |
| Hematologic disease, n (%) | 7 (1.5) | 0 |
| Immune disease, n (%) | 11 (2.3) | 2 (2.1) |
| Neurological disease, n (%) | 4 (0.84) | 0 |

[a]Participants answered the question "Have you been diagnosed by a physician with any of the following chronic medical conditions? (select all that apply)."
[b]Yr, year; SD, standard deviation; n, number; d, day.

Although efficacy of both the BNT162b2 (10) and mRNA-1273 (11) vaccines have been shown to remain > 95% against severe disease to at least 6 months, there are concerns that the effectiveness of the BNT162b2 vaccine may decrease with time against all-severity COVID-19 (12). Our immunogenicity data are consistent with a recent study demonstrating higher effectiveness of two doses of mRNA-1273, compared with two doses of BNT162b2, when Delta variant predominates (13). Collectively, these findings demonstrate higher magnitude 6-month humeral immunity with the mRNA-1273 vaccine, which may be explained by the higher relative mRNA dose (100 $\mu$g in the mRNA-1273 vaccine versus 30 $\mu$g in the BNT162b2 vaccine). Phase 1

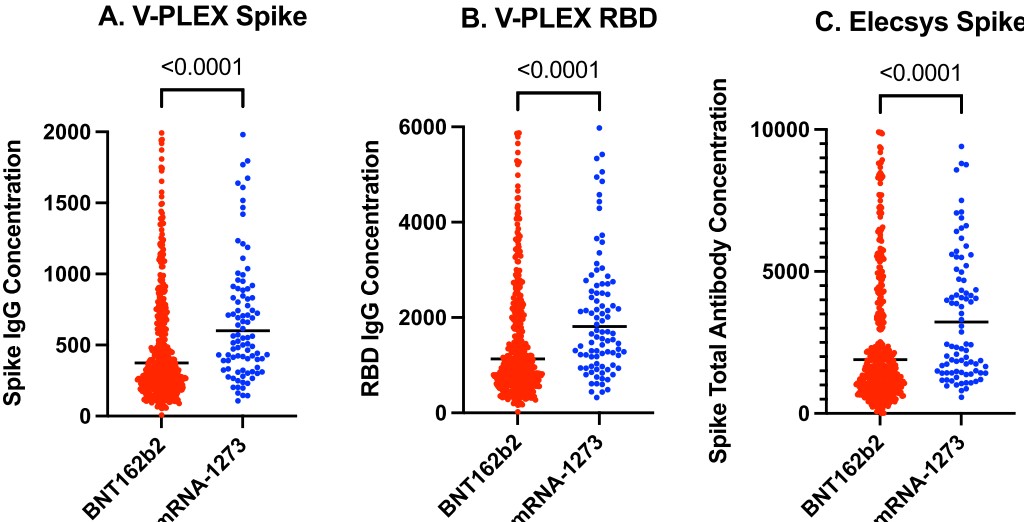

**FIG 1** Comparison of spike and receptor-binding domain antibody concentrations (BAU/mL), stratified by mRNA vaccine type. (A) Spike IgG antibody concentrations (BAU/mL), measured on the V-PLEX assay. (B) Receptor-binding domain (RBD) IgG antibody concentrations (BAU/mL), measured on the V-PLEX assay. (C) Spike total antibody concentrations (BAU/mL), measured on the Elecsys assay. The black line represents the geometric mean. P values derived from the Mann-Whitney U test. BAU/mL, binding antibody units per mL.

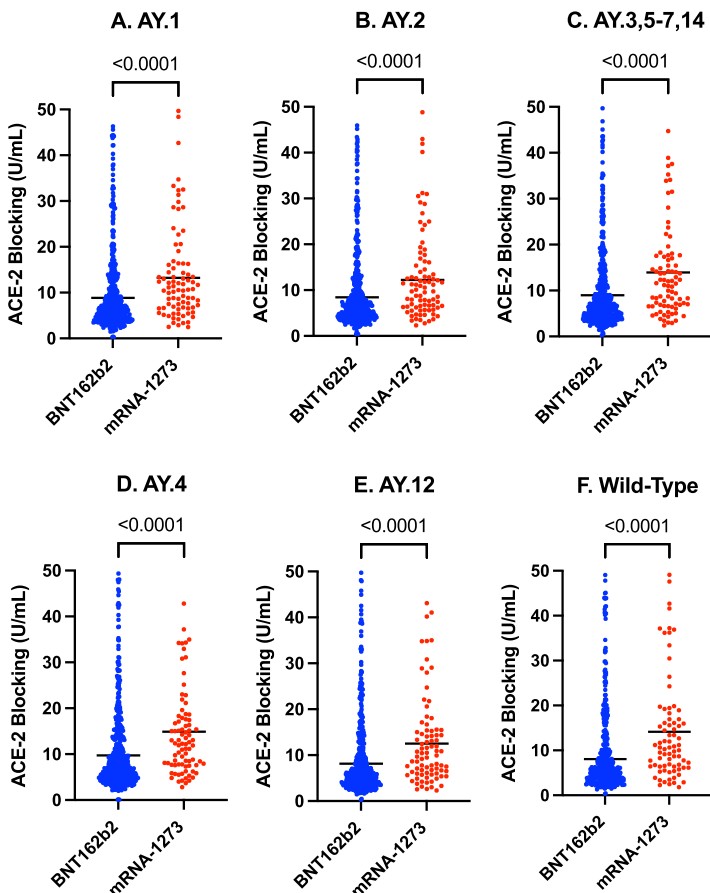

**FIG 2** Comparison of inhibition of ACE-2 binding to Delta variant and wild-type strain spike protein concentrations (U/mL), stratified by mRNA vaccine type. Delta variant spike proteins: A. AY.1.; B. AY.2.; C. B.1.617.2/AY.3/AY.5/AY.6/AY.7/AY.14; D. B.1.617.2/AY.4.; E. AY.12. Wild-Type Spike Protein results in Panel F. The black line represents the geometric mean. *P* values derived from the Mann-Whitney U test. U/mL, units per mL.

and 2 trials of the mRNA-1273 vaccine reported that a mRNA dose of 100 $\mu$g demonstrated higher immunogenicity than a 25 $\mu$g dose, but not in comparison with a 50 $\mu$g dose (14).

Given ongoing global supply constraints, particularly of more efficacious mRNA vaccines, and that close to half of the global population has yet to be vaccinated, long-term immunogenicity should be considered when choosing vaccine types, especially for remote and/or marginalized populations where administration of third doses may prove challenging. These data may also have implications when contemplating timing of third vaccine doses for the general population, to prioritize individuals vaccinated initially with BNT162b2 (vs. mRNA-1273) vaccines.

Previous studies have compared immunogenicity elicited from the Delta versus wild-type strains, with many showing a decrease in neutralization (7, 15). While we did not directly compare immunogenicity between SARS-CoV-2 strain-based groups, ACE-2 inhibition from the wild-type and all Delta lineages were similar, as were comparisons between vaccine types within each group. These findings are congruent with previous work showing similar neutralization sensitivity of the Delta variant and its sublineages (8, 9). Overall, our results indicate that the mRNA-1273 vaccine similarly elicits a higher immune response for the wild-type and all tested Delta strains.

Our study is subject to several limitations. These data include individuals with 7-week (on average) intervals between first and second doses which may partially explain relatively high concentrations of 6-month antibodies and may differ if

manufacturer recommended dosing schedules are used. Our study reports markers of humoral immune response, rather than clinical outcomes, which may not reflect all elements of protection and may not reflect clinical outcomes. We included middle-aged adult paramedics, whose immunity may differ from other groups. This was an observational study, which resulted in an uneven number of participants in the comparison groups. Conclusions are limited to association and confounders may have affected results. While the Delta variant accounted for over 99% of global COVID-19 at the time of this analysis (5), additional testing may be required to determine generalizability of these findings to new SARS-CoV-2 variants of concern such as Omicron.

In conclusion, we found that adults who received two doses of mRNA-1273 vaccines demonstrated higher wild-type and Delta variant-specific immunogenicity at 6 months, compared with those who received two doses of the BNT162b2 vaccine

## MATERIALS AND METHODS

**Parent study, study design, and setting.** The COVID-19 Occupational Risks, Seroprevalence and Immunity among Paramedics in Canada (CORSIP) study is a prospective observational cohort study of adult ($\geq$ 19 years of age) paramedics working in the provinces of British Columbia, Alberta, Saskatchewan, Manitoba, and Ontario. The CORSIP study was approved by the University of British Columbia (H2O-03620) and University of Toronto (40435) research ethics boards and started enrolling participants in January 2021, with written consent. Participants completed sociodemographic and health questionnaires detailing vaccination status and history of nucleic acid amplification test (NAAT)-confirmed COVID-19 infections. Participants were requested to provide a blood sample at enrollment, and also 6 months after the first vaccine dose (if applicable). Data from this study can be obtained from the COVID-19 Immunity Task Force.

**Selection of participants.** For this analysis, we included participants who received two doses of the BNT162b2 vaccine or two doses of the mRNA-1273 vaccine, and provided a blood sample 170 to 190 days after the first vaccine dose. We excluded individuals with previous COVID-19, defined as a positive NAAT test or a reactive test on the Elecsys Anti-SARS-CoV-2 nucleocapsid (Roche, IND, USA; see supplemental materials) assay (which is indicative of prior SARS-CoV-2 infection rather than vaccination).

**Outcome measures.** We used several assays in this investigation to evaluate immunogenicity, reporting antibody and inhibition of angiotensin-converting enzyme 2 (ACE-2) binding concentrations to SARS-CoV-2 antigens. While SARS-CoV-2 live viral neutralization testing is typically viewed as the optimal surrogate measure of immune protection—as has been associated with COVID-19 severity (16–19)—neutralization testing procedures are labor-intensive, precluding large volume testing. Alternatively, ACE-2 inhibition and anti-spike antibody concentrations have been shown to strongly correlate with live virus neutralization assays (20–23).

The primary outcome in this study was spike IgG antibody concentrations from the prototypic wild-type SARS-CoV-2 strain, measured with the V-PLEX COVID-19 Panel 2 IgG assay (Meso Scale Discovery, MD, USA), reported in the standardized "binding antibody units" per milliliter (BAU/mL) (24). Secondary outcomes included: IgG antibody concentrations against the RBD from the wild-type SARS-CoV-2 strain, measured with the V-PLEX COVID-19 Panel 2 IgG assay (Meso Scale Discovery, MD, USA; reported in BAU/mL); wild-type spike total antibody concentrations measured with the Elecsys Anti-SARS-CoV-2 S total antibody assay (Roche, IND, USA; reported in BAU/mL); ACE-2 binding to the wild-type spike protein, and AY.1, AY.2, B.1.617.2/AY.3/AY.5/AY.6/AY.7/AY.14, B.1.617.2/AY.4, and AY.12 spike proteins from Delta variant subtypes (6), measured on the V-PLEX SARS-CoV-2 Panel 19 ACE2 Kit (Meso Scale Discovery, MD, USA; measured in units/mL [U/mL]). Delta proteins were chosen for inclusion in Panel 19 ACE2 Kit based on the predominant lineages, however do not represent all Delta lineage strains (6).

**Statistical analyses.** We performed analyses using GraphPad Prism Version 9.2.0 (GraphPad Software, San Diego, CA). For participant characteristics, we described categorical variables as counts (with percentages) and continuous variables as means (with standard deviation [SD]). We reported outcomes as geometric mean (GM) with geometric standard deviation (GSD), and compared using the Mann-Whitney U test. A 2-sided $P$ value of $<0.05$ was classified as statistically significant.

## SUPPLEMENTAL MATERIAL

Supplemental material is available online only.
**SUPPLEMENTAL FILE 1**, PDF file, 0.1 MB.

## ACKNOWLEDGMENTS

We acknowledge the contributions of all the paramedics who participated in our study, as well as the contributions of Tara Martin, Bethany Poon, Heba Qazilbash, Paul Demers, Tracy Kirkham, Christopher MacDonald, Dong Vo, Yann-Charles Lafontant, David O'Neill, Jeff Maxim, Cheryl Cameron, Troy Clifford and Ambulance Paramedics of BC, Dave Deines and the National Paramedics Association of Canada, Justin Yap, Veronica Chow, Ryan Sneath, Jennifer Bolster, Nechelle Wall, Brian Twaites, Sandra Jenneson, Robert Schlamp, Ashley Curtis, Chelsie Osmond, Suzanne Vercauteren and

the BC Children's Hospital Biobank, and other participating paramedic services and unions.

S.D. has acted as a content expert for respiratory viruses for Johnson & Johnson (Janssen).

This study was supported by funding from Government of Canada, through the COVID-19 Immunity Task Force.

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
