## [Reviewer comments · Microbiology Spectrum]

Microbiology Spectrum

Comparative 6-month Wild-Type and Delta-Variant Antibody levels and Surrogate Neutralization for Adults Vaccinated with BNT162b2 vs. mRNA-1273

Brian Grunau, Liam Golding, Martin Prusinkiewicz, Michael Asamoah-Boaheng, Richard Armour, Ana Marquez, Agatha Jassem, Vilte Barakauskas, Sheila O'Brien, Steven Drews, Scott Haig, Pascal Lavoie, and David Goldfarb

Corresponding Author(s): Brian Grunau, University of British Columbia

Review Timeline:

Submission Date:	December 21, 2021
Editorial Decision:	January 17, 2022
Revision Received:	January 19, 2022
Accepted:	February 13, 2022

Editor: Rafael A. Medina

Reviewer(s): Disclosure of reviewer identity is with reference to reviewer comments included in decision letter(s). The following individuals involved in review of your submission have agreed to reveal their identity: Kai Dallmeier (Reviewer #2)

Transaction Report:

DOI: <https://doi.org/10.1128/spectrum.02702-21>

January 17, 2022

Dr. Brian Grunau
University of British Columbia
Department of Emergency Medicine
Vancouver, British Columbia
Canada

Re: Spectrum02702-21 (Comparative 6-month Delta Variant Surrogate Neutralization and Antibody levels for Adults Vaccinated with BNT162b2 vs. mRNA-1273)

Dear Dr. Brian Grunau:

Thank you for submitting your manuscript to Microbiology Spectrum. You will see from the referees' comments that additional information needs to be provided for clarity. Please, provide a more elaborate introduction section to properly introduce the scope of the study. Also a more detailed and critical description of the methodology would be valuable.

As you will see your manuscript is very close to acceptance. Please modify the manuscript along the lines I have recommended. As these revisions are quite minor, I expect that you should be able to turn in the revised paper in less than 30 days, if not sooner. If your manuscript was reviewed, you will find the reviewers' comments below.

When submitting the revised version of your paper, please provide (1) point-by-point responses to the issues I raised in your cover letter, and (2) a PDF file that indicates the changes from the original submission (by highlighting or underlining the changes) as file type "Marked Up Manuscript - For Review Only". Please use this link to submit your revised manuscript. Detailed instructions on submitting your revised paper are below.

Link Not Available

Sincerely,

Rafael A. Medina

Reviewer comments:

Reviewer #1 (Comments for the Author):

In this work, authors present comparison of the efficacy of the two mRNA vaccines against COVID-19, namely BNT162b2 and mRNA-1273, at approx. 6-months after the first dose. Since the majority of the world population was infected with the Delta variant of SARS-CoV-2 at the time of the preparation of the manuscript, the authors are specially focused on this variant. The method for testing vaccine efficacy was ACE-2 binding inhibition as a surrogate to the traditional virus neutralization assays. The results clearly show the Spike protein-specific humoral immune response to be significantly higher in mRNA-1273-vaccinated individuals than in the BNT162b2-vaccinated individuals for all the Spike variants.

The authors discuss nicely the limitations of their work especially the fact that they focus merely on the humoral immune response to Spike protein which may not completely correlate with the clinical outcome.

This is an interesting and valuable work showing a potential of mRNA COVID-19 vaccines. A follow up of this study would greatly improve our knowledge about the long-term efficacy and cross-reactivity of these two vaccines.

The manuscript is well written, methods are presented clearly, experiments and analyses were performed correctly. Also, the claims are supported by adequate references.

Minor comment:

The authors should discuss disbalance between the numbers of participants receiving either vaccine (475 vs. 96). Why the numbers of participants are so different and could this have any influence on the final conclusion?

Reviewer #2 (Comments for the Author):

Grunau and colleagues provide a comprehensive data set on a clinical COVID19 vaccine cohort, comparing the immunogenicity of two related mRNA vaccines. The results are largely descriptive and fully sound. The paper elegantly written and very readable.

The introduction may benefit from a more elaborate narrative framing the importance/justification of the study. It may be in particular interesting to give some more background on the fairly unique broad assessment of the full spectrum of Delta virus subtypes performed (Why was this desirable? Are there clinical consequences? What is known about the necessity to subtype? Are there serotypes at all?). Finally, more a detailed and critical description of the methodology (approach and limits) may be required. This appears essential to me, as the entire study is based basically on a single assay principle that is not further validated.

In the results section, the authors may consider a shift in the order of the two main figures. Not required for the scope of the study, but available in the data (yet not displayed), a pairwise correlation of prototype and Delta-specific antibody responses may help to understand the impact of the antigenic drift in current virus variants may have for vaccine efficacy (revealing any shift in GMT titers or equivalent ACE2-binding inhibition titers), as documented by others. Expression of antibody titers in International Units may be desirable; alternatively, the used standards may some need more explanation to allow comparison with other studies.

More details and specific comments are provided in attached PDF.

Preparing Revision Guidelines

- point-by-point responses to the issues I raised in your cover letter
- Upload a compare copy of the manuscript (without figures) as a "Marked-Up Manuscript" file.
- Each figure must be uploaded as a separate file, and any multipanel figures must be assembled into one file.
- Manuscript: A .DOC version of the revised manuscript
- Figures: Editable, high-resolution, individual figure files are required at revision, TIFF or EPS files are preferred

Please return the manuscript within 60 days; if you cannot complete the modification within this time period, please contact me. If you do not wish to modify the manuscript and prefer to submit it to another journal, please notify me of your decision immediately so that the manuscript may be formally withdrawn from consideration by Microbiology Spectrum.

Comparative 6-month Delta Variant Surrogate Neutralization and Antibody levels for Adults Vaccinated with BNT162b2 vs. mRNA-1273

by Grunau et al.

In this work, authors present comparison of the efficacy of the two mRNA vaccines against COVID-19, namely BNT162b2 and mRNA-1273 at approx. 6-months after the first dose. Since the majority of the world population was infected with the Delta variant of SARS-CoV-2 at the time of the preparation of the manuscript, the authors are specially focused on this variant. The method for testing vaccine efficacy was ACE-2 binding inhibition as a surrogate to the traditional virus neutralization assays. The results clearly show that Spike protein-specific humoral immune response to be significantly higher in mRNA-1273-vaccinated individuals than in the BNT162b2-vaccinated individuals for all the Spike variants.

The authors discuss nicely the limitations of their work especially the fact that they focus merely on the humoral immune response to Spike protein which may not completely correlate with the clinical outcome.

This is an interesting and valuable work showing a potential of mRNA COVID-19 vaccines. A follow up of this study would greatly improve our knowledge about the long-term efficacy and cross-reactivity of these two vaccines.

The manuscript is well written, methods are presented clearly, experiments and analyses were performed correctly. Also, the claims are supported by adequate references.

Minor comment:

The authors should discuss disbalance between the numbers of participants receiving either vaccine (475 vs. 96). Why the numbers of participants are so different and could this have any influence on the final conclusion?

Reviewer #1 (Comments for the Author):

In this work, authors present comparison of the efficacy of the two mRNA vaccines against COVID-19, namely BNT162b2 and mRNA-1273, at approx. 6-months after the first dose. Since the majority of the world population was infected with the Delta variant of SARS-CoV-2 at the time of the preparation of the manuscript, the authors are specially focused on this variant. The method for testing vaccine efficacy was ACE-2 binding inhibition as a surrogate to the traditional virus neutralization assays. The results clearly show the Spike protein-specific humoral immune response to be significantly higher in mRNA-1273-vaccinated individuals than in the BNT162b2-vaccinated individuals for all the Spike variants.

The authors discuss nicely the limitations of their work especially the fact that they focus merely on the humoral immune response to Spike protein which may not completely correlate with the clinical outcome. This is an interesting and valuable work showing a potential of mRNA COVID-19 vaccines. A follow up of this study would greatly improve our knowledge about the long-term efficacy and cross-reactivity of these two vaccines.

The manuscript is well written, methods are presented clearly, experiments and analyses were performed correctly. Also, the claims are supported by adequate references.

Thank you for your review of our submission.

Minor comment:

The authors should discuss disbalance between the numbers of participants receiving either vaccine (475 vs. 96). Why the numbers of participants are so different and could this have any influence on the final conclusion?

Thank you for this comment (which was also mentioned by Reviewer 2 [manuscript comment #12]). This was an observational study and thus the data we collected reflects the vaccine administration provided to enrolled paramedics in our study. We did not have any influence on the types of vaccines received by our participants. The sample sizes in these groups impact the power of our study to see differences, however we estimate that differences in the comparison group sizes are unlikely to have affected the results. The characteristics in both groups are similar. We have added additional comments to the limitations section to describe these limitations: "This was an observational study, which resulted in an uneven number of participants in the comparison groups. Conclusions are limited to association and confounders may have affected results."

Reviewer #2 (Comments for the Author):

Grunau and colleagues provide a comprehensive data set on a clinical COVID19 vaccine cohort, comparing the immunogenicity of two related mRNA vaccines. The results are largely descriptive and fully sound. The paper elegantly written and very readable.

Thank you for your thoughtful review of our manuscript.

The introduction may benefit from a more elaborate narrative framing the importance/justification of the study. It may be in particular interesting to give some more background on the fairly unique broad assessment of the full spectrum of Delta virus subtypes performed (Why was this desirable? Are there clinical consequences? What is known about the necessity to subtype? Are there serotypes at all?).

Thank you for this suggestion. We have added additional background to the introduction.

Finally, more a detailed and critical description of the methodology (approach and limits) may be required. This appears essential to me, as the entire study is based basically on a single assay principle that is not further validated.

Thank you for this suggestion. We have inserted further description and rationale for using these outcome measures at the beginning of the "Outcome Measures" section. We have also added to the limitations section that the outcome measures used in this study "may not reflect clinical outcomes".

In the results section, the authors may consider a shift in the order of the two main figures.

Based on this suggestion, we have shifted the order of the two main figures. We have also shifted the primary outcome to correspond to this change in order.

Not required for the scope of the study, but available in the data (yet not displayed), a pairwise correlation of prototype and Delta-specific antibody responses may help to understand the impact of the antigenic drift in current virus variants may have for vaccine efficacy (revealing any shift in GMT titers or equivalent ACE2-binding inhibition titers), as documented by others.

Thank this suggestion for a follow-up analysis!

Expression of antibody titers in International Units may be desirable; alternatively, the used standards may some need more explanation to allow comparison with other studies.

Thank you for this suggestion. The recommended units by WHO for SARS-CoV-2 antibody assays is binding antibody units (BAU), and for neutralizing antibody activity is IU ([https://www.thelancet.com/journals/lanct/article/PIIS0140-6736\(21\)00527-4/fulltext](https://www.thelancet.com/journals/lanct/article/PIIS0140-6736(21)00527-4/fulltext), [https://www.thelancet.com/journals/lanmic/article/PIIS2666-5247\(21\)00266-4/fulltext](https://www.thelancet.com/journals/lanmic/article/PIIS2666-5247(21)00266-4/fulltext)). Based on your suggestion we have converted our antibody results to BAU's, and redid the analysis and created new figures. For ACE2 inhibition there is no recommended standardized reporting unit, and thus we have left as unit/mL,

The WHO has developed an international standard for SARS CoV2 serological assays to help compare results and outcomes of different studies. Both of these assays probably can have their results converted to BAU/mL (binding antibody units).

More details and specific comments are provided in attached PDF.

Thank you for these additional comments and suggestions to improve our manuscript. We have:

- We added additional details to the introduction to support the rationale for the study, including the rationale for examining multiple delta-variant lineages (manuscript comment #6). We have indicated that robust studies comparing clinical features of the different delta lineages is lacking.
- We added additional details and support in the Methods for the use of protein-based assays to measure immunogenicity, but also added to the limitations section that these may not reflect clinical outcomes (manuscript comments #3, 5, 7). We have added details on the types of antibodies measured in the different assays. The type of antibody measured by the V-PLEX assay is IgG. The Elecsys assay is a total antibody assay.
- We have added additional details to the introduction and discussion regarding differences between the various delta lineages. There are currently few data comparing outcomes between these strains, however we have commented on two studies in the discussion which reported that immunogenicity between different strains were similar (manuscript comment #4). We have also cited the cov-

lineages.org website, where readers can refer to for further details on all delta variant lineages and nomenclature

- The delta proteins included in the ACE-2 kit were chosen based on the predominant lineages at that time. It is not an exhaustive list (manuscript comment #8). We have added these details to the manuscript
- We have changed our references to the “Wuhan” strain, to the terms “prototypic” or “wild-type” (manuscript comment #10).
- Based on your suggestion, we have changed the primary outcome to the wild-type strain results, with delta-related results included as secondary outcomes (manuscript comment #13). As such, we have switched Figures 1 and 2 as suggested.
- We have added a fourth paragraph in the discussion commenting on differences (or lack thereof) between delta variants tested (manuscript comment #15).
- Regarding manuscript comment #17, there was some work done as part of the initial FDA submission regarding the immunogenicity of different doses of the mRNA-1273 vaccine. We have added these details to the 2nd paragraph of the discussion.
- Regarding manuscript comment #18, we agree that the word “persisting” is not correct, and thus we have amended this sentence. We do not have a prior sample for many of the participants in this study, and thus are not able to comment on kinetics.
- With regards to the other more minor suggestions in the manuscript comments, we have made edits based on your suggestions.

February 13, 2022

Dr. Brian Grunau
University of British Columbia
Department of Emergency Medicine
Vancouver, British Columbia
Canada

Re: Spectrum02702-21R1 (Comparative 6-month Wild-Type and Delta-Variant Antibody levels and Surrogate Neutralization for Adults Vaccinated with BNT162b2 vs. mRNA-1273)

Dear Dr. Brian Grunau:

We appreciate your consideration of the reviewers' comments and for the submission of a revised version.

Your manuscript has been accepted, and I am forwarding it to the ASM Journals Department for publication. You will be notified when your proofs are ready to be viewed.

Sincerely,

Rafael A. Medina
Editor, Microbiology Spectrum
